# The Role of Vitamin A in Retinal Diseases

**DOI:** 10.3390/ijms23031014

**Published:** 2022-01-18

**Authors:** Jana Sajovic, Andrej Meglič, Damjan Glavač, Špela Markelj, Marko Hawlina, Ana Fakin

**Affiliations:** 1Eye Hospital, University Medical Centre Ljubljana, Grablovičeva 46, 1000 Ljubljana, Slovenia; jana.sajovic@gmail.com (J.S.); andrej.meglic.info@gmail.com (A.M.); spela.markelj@kclj.si (Š.M.); marko.hawlina@gmail.com (M.H.); 2Department of Molecular Genetics, Institute of Pathology, Faculty of Medicine, University of Ljubljana, Vrazov trg 2, 1000 Ljubljana, Slovenia; damjan.glavac@mf.uni-lj.si

**Keywords:** vitamin A, visual cycle, retinal diseases, ABCA4, RHO, RDH5, RDH12, treatment

## Abstract

Vitamin A is an essential fat-soluble vitamin that occurs in various chemical forms. It is essential for several physiological processes. Either hyper- or hypovitaminosis can be harmful. One of the most important vitamin A functions is its involvement in visual phototransduction, where it serves as the crucial part of photopigment, the first molecule in the process of transforming photons of light into electrical signals. In this process, large quantities of vitamin A in the form of 11-*cis*-retinal are being isomerized to all-*trans*-retinal and then quickly recycled back to 11-*cis*-retinal. Complex machinery of transporters and enzymes is involved in this process (i.e., the visual cycle). Any fault in the machinery may not only reduce the efficiency of visual detection but also cause the accumulation of toxic chemicals in the retina. This review provides a comprehensive overview of diseases that are directly or indirectly connected with vitamin A pathways in the retina. It includes the pathophysiological background and clinical presentation of each disease and summarizes the already existing therapeutic and prospective interventions.

## 1. Introduction

Vitamin A is a group of unsaturated organic compounds which is mostly associated with retinol. It occurs in various forms. Vitamin A (retinol), its natural (retinal, all-*trans* retinoic acid (tretinoin), 9-*cis*-retinoic acid (alitretinoin) and 13-*cis*-retinoic acid (isotretinoin)) and synthetic (etretinate, acitretin, adapalene, tazarotene and bexarotene, trifarotene) derivates are termed retinoids [1]. Different forms of vitamin A can convert from one form to another with the help of specific enzymes. The two major biologically active forms are 11-*cis*-retinal and all-*trans*-retinoic acid. All-*trans*-retinoic acid is an essential signalling molecule in cell differentiation, proliferation and apoptosis, and participates in gene transcription. It is crucial for reproduction, growth and development of the embryo, normal brain function, immune system functioning, skeletal development, growth of hair and nails, and epithelial cell RNA synthesis including the conjunctiva and cornea. The primary role of 11-*cis*-retinal is visual perception [1,2].

There are specific biochemical pathways involved in vitamin A transfer to and storage in the retinal pigment epithelium (RPE), transfer to the photoreceptors, conversion to its active form (11-*cis*-retinal), recycling of the inactive forms and removal of the toxic by-products; which will be reviewed in the 1st part of this paper. Pathogenic variants in genes, encoding the proteins involved in these processes, have been implicated in several retinal diseases, which will be reviewed in 2nd part of this paper.

### 1.1. Food Sources of Vitamin A

Vitamin A is an essential micronutrient, which means that the human body cannot synthesize it on its own and must be ingested. However, as it is one of the fat-soluble vitamins, it is stored in the body, in the form of retinyl ester, it does not need to be added daily. The highest amounts of vitamin A are stored in the liver, but it is also found in adipose tissue, lungs and the RPE [2,3,4,5,6].

Retinol and retinyl ester are the most common forms of vitamin A in food of animal origin, e.g., dairy products, eggs, fish and meat, especially liver. In plant-based foods, the main sources of vitamin A are beta-carotene, alpha-carotene and beta-cryptoxanthin. They are so called provitamin A carotenoids, as they are vitamin A precursors, that convert to vitamin A in the body. Depending on the body’s needs, only a certain proportion of ingested provitamins are metabolized to vitamin A, while the rest remain unchanged and serve as antioxidants. Antioxidant properties are also found in non-provitamin A carotenoids. The key representatives of the latter are lutein, zeaxanthin and lycopene [1,7,8,9].

Carotenoids are pigments that give fruits and vegetables red, orange and yellow hues. We can find them in sweet potatoes, carrots, melons, peppers, apricots, nectarines, tomatoes, mangoes and corn. They are also present in spinach, peas, parsley, pistachios, kale, broccoli and Swiss chard, where green chlorophyll masks the carotenoid pigment [1,10].

The recommended daily dose of vitamin A is around 400–500 µg for children and infants, 900 µg for adult men and 700 µg for adult women [1], while lactating women require up to 1300 µg per day [11]. In adults, stores of vitamin A in the body can last around 1 to 2 years while in children stores last much shorter and signs of deficiency develop more rapidly [1].

There are several analytical approaches to quantify different vitamin A metabolites levels in tissues and biological fluids. Their concentrations reflect vitamin A status and provide us with an insight into its metabolism, homeostasis and help us to overcome pathological conditions associated with altered vitamin A metabolism [1]. Excess (hypervitaminosis) or deficiency (hypovitaminosis) of vitamin A can both be harmful [1,10,12].

### 1.2. Hypervitaminosis A

Hypervitaminosis A can occur due to taking oral preparations which contain vitamin A, or high amounts of vitamin A rich food sources, such as liver. The disease can be acute if very high doses are ingested in a short time, or chronic, if the ingestion of excessive doses occurs over a longer period. In the acute form, the symptoms appear a few hours after the intake, and may include dizziness, disturbances of consciousness, irritability, nose bleeding, headache, nausea, vomiting, and abdominal pain [9,10,13]. Early signs of chronic hypervitaminosis include thinning of hair and eyebrows, cracked lips, and dry and rough skin. With time, sleep disturbances, blurred vision, photosensitivity, reduced appetite, weight loss, headaches, weakness, and bone pain may appear. Patients can have increased intracranial pressure and hepatosplenomegaly. Pregnant women must be cautious when taking vitamin A supplements, as high levels of vitamin A are teratogenic and can cause foetal defects in the developing foetus [1,12,13].

Retinoids, oral and topical, are frequently used to treat different skin disorders, e.g., psoriasis, photodamage, seborrhoea, acne, and ichthyosis. Isotretinoin is usually recommended in severe acne, as it has anti-inflammatory, anti-bacterial and anti-seborrheic properties. It is only used orally and is the most commonly used oral retinoid in dermatology [1,14,15]. In retina in the retinal pigment epithelium (RPE), isotretinoin inhibits two key visual cycle enzymes, RPE-specific 65 kDa protein (RPE65) [16] and 11-*cis*-retinol dehydrogenase [17,18] (Figure 1). This impairs chromophore, 11-*cis*-retinal, and slows rhodopsin (RHO) regeneration which can cause night blindness. However, on the other hand, it reduces the accumulation of toxic byproducts of the visual cycle, such as *N-retinyl-N-retinylidene ethanolamine* (A2E) [16,17,18]. Topical retinoids cause local side effects with skin erythema and peeling as the most common [15].

Carotenoids (precursors of vitamin A) are considered relatively safe, and their toxicity is extremely rare. Excessive and prolonged beta-carotene consumption causes yellow-orange discolouration of skin thick areas, such as palms and soles, however this is a benign, reversible condition [1,19]. Nevertheless, adverse health effects concerning carotenoids have been reported in smokers who had a higher incidence of lung cancer and mortality rate after high doses of beta-carotene [1,10,20,21,22]. Therefore, for treatment of age-related macular degeneration (AMD), the original Age-Related Eye Disease Study (AREDS) formulation that included beta-carotene was replaced with the AREDS2 formula, where beta-carotene was substituted with lutein and zeaxanthin [23].

### 1.3. Hypovitaminosis A

Clinical signs of hypovitaminosis A depend on the patient’s age and the severity of the vitamin A deficiency. Infants and preschool children are at the highest risk due to low vitamin A stores at birth and high body growth needs. Therefore, they may suffer from growth delay [10]. Hypovitaminosis A affects all tissues, but the most characteristic signs involve the skin and eyes.

Dermatologic manifestations include generalized xerosis and follicular hyperkeratosis, with patients exhibiting dry, rough and peeling skin [1,10,12,24]. Ocular symptoms include severe dry eye with conjunctival and corneal xerosis and Bitot’s spots, as well as night blindness, which will be described in detail below [25].

The majority of preformed vitamin A consists of retinyl esters [1,26]. Dietary retinyl esters are first hydrolysed to retinol via the actions of pancreatic lipase in the duodenum and phospholipase B at the brush border of enterocytes. Retinol is then absorbed by the enterocyte, where it is re-esterified to retinyl ester [1,2,26,27,28,29]. Absorption of retinol is increased if consumed with fatty meals [1,27,28]. In the enterocyte formed retinyl esters are packaged in nascent chylomicrons that are secreted into the lymphatic system for uptake into the body [1,2,26,28,29,30,31]. Dietary provitamin A carotenoids are taken into the enterocyte by scavenger receptors where they can either be converted to retinyl esters or packaged unmodified into nascent chylomicrons [2,27,28]. After traversing the lymphatic system, chylomicrons enter the bloodstream and are then mainly delivered to the liver, where they are stored mostly in the form of retinyl esters, predominately as retinyl palmitate, and stearate [28,32,33]. The liver is the main storage organ of vitamin A in the body, as up to 80% of vitamin A in the body is stored there [2,13,28,30,31]. When needed, retinyl esters are hydrolysed into retinol. Its mobilisation to extrahepatic tissues requires retinol-binding protein (RBP4), the main vitamin A carrier in the blood. In the liver, retinol and RBP4 (apo-RBP4) form the complex (holo-RBP4), which is then secreted into the bloodstream in order to meet peripheral tissue needs [1,2,26,28,29,30,31].

Any condition that interferes with ingestion (e.g., malnutrition), absorption (e.g., Crohn disease, pancreatic insufficiency), storage (e.g., alcoholic cirrhosis), or transport of vitamin A (e.g., pathogenic variants in *RBP4* gene) can lead to a deficiency of vitamin A in the target tissues. Diseases causing compromised fat malabsorption (e.g., cystic fibrosis, coeliac disease) are also a cause of vitamin A deficiency [12,33,34].

### 1.4. Vitamin A Pathways in the Retina

#### 1.4.1. Transport from Blood Circulation to the Photoreceptor Outer Segments

The key transporters and enzymes involved in vitamin A pathways in the retina are summarized in the Figure 1 and Table 1. Vitamin A is a fat-soluble molecule and is dependent on water-soluble carriers. In the blood circulation, it is transported in the form of all-*trans*-retinol bound to a 21 kDa carrier plasma retinol-binding protein 4 (RBP4). To prevent loss of the low molecular weight RBP4 through renal filtration, holo-RBP4 forms a complex with a 55 kDa transthyretin (TTR), that is also known for transporting thyroxine [35,36,37]. The complex holo-RBP4:TTR diffuses from the choroidal blood circulation through the Bruch’s membrane and binds to the RBP4 receptor on the RPE cells; named stimulated by retinoic acid 6 (STRA6) [38,39]. Transthyretin blocks the ability of holo-RBP4 to associate with STRA6, and, therefore, needs to dissociate from holo-RBP4:TTR complex beforehand [37]. The STRA6 receptor facilitates the uptake of all-*trans*-retinol to the cytoplasmic side, where it associates with another carrier, a 15 kDa cellular retinol-binding protein 1 (CRBP1) [40,41,42,43]. Inside the RPE, CRBP1 shuttles all-*trans*-retinol to the enzyme retinol:lecithin acyltransferase (LRAT) which converts all-*trans*-retinol to all-*trans* retinyl esters [44,45]. Retinyl esters can either accumulate and store in the retinosomes, specific lipid droplet-like storage particles in the eyes [4], or represent substrates for the key enzyme in the visual cycle, RPE65. RPE65 is an isomerohydrolase that catalyses hydrolysis of esters and isomerizes all-*trans*-retinol to 11-*cis*-retinol [29,32,46]. 11-*cis*-retinol is then bound by a cellular retinaldehyde-binding protein (CRALBP), which prevents unwanted isomerisation to all-*trans*-retinol and further esterification to retinyl esters. CRALBP also facilitates oxidation of 11-*cis*-retinol to 11-*cis*-retinal, the molecule in visual transduction, by 11-*cis*-retinol dehydrogenases, 11-*cis*-retinol dehydrogenase 5 (RDH5) and 11-*cis*-retinol dehydrogenase 11 (RDH11) [47,48,49]. 11-*cis*-retinal is then released into the interphotoreceptor matrix, where it binds interphotoreceptor retinoid-binding protein (IRBP), which delivers it to the outer segments of photoreceptors. There 11-*cis*-retinal combines with opsin to form visual pigment [47,50,51] (Figure 1). The visual pigments in rods and cones are rhodopsin (RHO) and iodopsin, respectively. The structure of rod and cone outer segments differs in a way that affects the localization of the visual pigment. In rods, the visual pigment is located on membranes of stacked discs inside the outer segment, whereas in cones the discs (also called lamellae) are partly fused with the plasmalemma, and thus in contact with extracellular space [52,53,54] (Figure 1).

#### 1.4.2. Phototransduction

Phototransduction is a process by which light is converted to electrical signals. The first step in the process is absorption of photons by the visual pigment, which causes isomerization of 11-*cis*-retinal to all-*trans*-retinal. This change triggers further biochemical reactions downstream that results in the change of the electrical potential across the photoreceptor membrane. Electrical signals from the retina then travel through the optic nerve to the brain, where they are converted into an image [29].

#### 1.4.3. The Visual Cycle

The all-*trans*-retinal is continuously recycled back to the 11-*cis*-retinal in the process called the visual cycle. The classical, or canonical visual cycle involves regeneration of 11-*cis*-retinal in the RPE. When the all-*trans*-retinal is released from the opsin, on the luminal side of the photoreceptor membrane of rod discs and extracellular side in cones, it binds to the phosphatidylethanolamine to form N-retinylidene-phosphatidylethanolamine. The latter is transported by the ATP-binding cassette subfamily A member 4 transporter (ABCA4) across the membrane to the cytoplasmic side of the photoreceptor. All-*trans*-retinal is released into the cytoplasm, where it can be converted to all-*trans*-retinol by all-*trans*-retinol dehydrogenases, RDH8 in the outer segments of the photoreceptors and RDH12 in the inner segments of the photoreceptors. The IRBP helps transfer all-*trans*-retinol from photoreceptors to the RPE, where it can be converted back to 11-*cis*-retinal by the LRAT, RPE65, RDH5 and RDH11 enzymes as described above. Regenerated 11-*cis*-retinal is then again available to form the visual pigment for phototransduction [29,32,46]. In RPE, an additional source of 11-*cis*-retinal is also a pathway with retinal G protein-coupled receptor (RGR) as the key enzyme. RGR is a RHO homologue, which, contrary to RHO, preferentially binds all-*trans*-retinal. Within RGR, a photon of light converts all-*trans*-retinal to 11-*cis*-retinal, the chromophore of the visual pigments. Therefore, RGR, together with RPE65, contributes to the regeneration of 11-*cis*-retinal [55,56,57,58].

There is a second, so called cone visual cycle, that occurs in the Müller cells. It serves cones with fast recycling, necessary for their rapid dark adaptation and function in constant and bright illumination [29,47,59,60,61,62]. The key proteins and their functions in this pathway remain unclear. It has been proposed, that all-*trans*-retinol is from cones transported by IRBP to Müller cells where it is isomerized to 11-*cis*-retinol by the dihydroceramide desaturase-1 (DES1) [47,63,64]. DES1 physically associates with CRALBP, which binds 11-*cis*-retinol, prevents its further isomerization and releases it into the interphotoreceptor matrix, to bind with IRBP [63]. IRBP then transfers 11-*cis*-retinol to cone inner segments, from where it is transported with undefined mechanism to cone outer segments. There unknown 11-*cis*-RDH oxidizes it to 11-*cis*-retinal needed for visual pigment regeneration [65]. As rods depend entirely on 11-*cis*-retinal and cones can also utilize 11-*cis*-retinol, the competition for this visual cycle product is avoided [52,63,65,66]. It is proposed, that 11-*cis*-retinol for cones is in Müller cells regenerated also by the complex of retinol dehydrogenase 10 (RDH10) and RGR opsin [67] (not shown on Figure 1).

#### 1.4.4. Accumulation and Clearance of Toxic Derivatives of Vitamin A

Cone and rod outer segments are renewed once every 10 days [68]. Through this process, a new membrane is added at the base of the outer segments while aged outer segment ends are removed by shedding and phagocytosis by adjacent RPE cells [69].

In the membranes of outer photoreceptor segments are nonenzymatically formed toxic bisretinoids. Upon phagocytosis, these bisretinoids are in the acidic environment of RPE phagolysosomes converted to bisretinoid pyridinium compound A2E and related bisretinoids [54,69,70,71,72,73,74]. As A2E can not be further metabolized and as RPE cells are postmitotic, A2E and related bisretinoids accumulate in the RPE in the form of lipofuscin deposits. Lipofuscin is an autofluorescent cellular waste product that compromises the function and viability of RPE cells and ultimately results in photoreceptor degeneration and a loss of vision [54,69,73,74,75].

Although abundant lipofuscin accumulation is involved in the aetiology of certain retinal diseases (e.g., AMD, Best disease and ABCA4-retinopathy), it accumulates at a lower level also with age in the RPE of healthy eyes [9,72,73,74,75].

## 2. Retinal Diseases Directly or Indirectly Associated with Vitamin A and Its Pathways

Vitamin A is needed for normal visual function and vitamin A deficiency (hypovitaminosis A) results in visual dysfunction that is reversible with vitamin A supplementation.

Several retinal diseases are directly or indirectly associated with vitamin A. Some are caused by pathogenic variants in genes encoding proteins that are directly involved in the vitamin A pathways in the retina, leading to local hypovitaminosis and/or accumulation of toxic bisretinoids. Other involve visual dysfunction due malfunction of the visual pigment or local deficiency of vitamin A due to indirectly impaired transport of vitamin A from choroid to the retina. Treatment with vitamin A in these cases is not straight-forward, as there is a fear of accumulation of excessive amounts of toxic retinoids that are formed as byproducts of vitamin A pathways. A good example of this toxicity is ABCA4-retinopathy where toxic products of vitamin A accumulate excessively in the retina [9,23,76,77,78].

### 2.1. Retinal Signs of Hypovitaminosis A

Vitamin A deficiency in the retina primarily affects the rods, which results in night blindness, followed by cone dysfunction and impairment of daytime vision, including visual acuity. The delayed impairment of cones is thought to be due to their additional pathway for production of 11-*cis*-retinal in Müller cells [79].

The functional impairment may be demonstrated using electrophysiology [80,81,82]. Interestingly, S-cones were shown to be affected prior to the L and M cones [81]. Recently, optical coherence tomography (OCT) has been shown as a useful diagnostic method, which shows abnormalities at the level of photoreceptor outer segments [83] (Figure 2). Similarly, vitamin A deficiency can be monitored by fundus autofluorescence (FAF) imaging. It may reveal hypoautofluorescent lesions [84,85] and decreased autofluorescence signal [86,87,88]. Reduced FAF can also be found in retinal diseases with impaired function of enzymes, involved in the conversion of all-*trans*-retinol into 11-*cis*-retinal in RPE cells. These enzymes include RPE65, LRAT, RLBP1, RDH5 and RDH11 [87,89]. Vitamin A replacement results in improvement of retinal function as well as structure, however long-term deficiency can lead to permanent degeneration of photoreceptors [31,80,81,83,84,85,86,87,88,90]. Interestingly, hypovitaminosis A has also been associated with reversible ganglion cell thinning [83], the reason for which is not understood but may reflect other vitamin A roles in the retina.

### 2.2. Retinal Diseases Associated with Pathogenic Variants in Genes Encoding Proteins Involved with Visual Cycle or Phototransduction

#### 2.2.1. ABCA4-Retinopathy

Stargardt disease (STGD1), also known as fundus flavimaculatus or ABCA4-retinopathy, is a progressive disorder of the retina caused by bi-allelic variants in the gene that encodes ATP-binding cassette subfamily A member 4 (ABCA4), localized on chromosome 1p22.1. It is the most frequent retinal dystrophy caused by a single gene and is believed to affect approximately 1 in 8000–10,000 individuals worldwide [91]. ABCA4 protein is a 250 kDa flippase, which transports all-*trans*-retinal, across the photoreceptor membrane to the cytoplasmic side, making it an important part of the visual cycle and removal of toxic vitamin A products [71,92].

ABCA4 dysfunction leads to the accumulation of vitamin A derivatives and toxic compounds, as they cannot be efficiently removed from the photoreceptor outer segment. Trapped vitamin A derivatives and toxic compounds can react nonenzymatically, leading to the formation of toxic bisretinoids. Upon phagocytosis of outer segments by adjacent RPE cells, the bisretinal compounds can be hydrolyzed by lysosomal enzymes to highly toxic metabolite A2E. The latter are insoluble, have a tendency to aggregate and accumulate as lipofuscin deposits in the RPE cells, compromising RPE function and leading to cell death. Consequently, dysfunction and loss of photoreceptors they support occurs [69,71,93].

In STGD1, cone degeneration is more severe and occurs before rod degeneration [69,94]. The reason for that is not completely understood, however it has been proposed that direct cone toxicity also plays a role in the disease pathogenesis, potentially due to the cone open lamellae [94]. There are currently >1290 known pathogenic variants in the *ABCA4* gene (www.lovd.nl/ABCA4, accessed on 18 October 2021), causing a heterogeneous phenotypic appearance. Typically, the disease affects central vision, however, in more severe forms, it can also lead to blindness [95]. Onset is most commonly in childhood or adolescence and least frequently in later adulthood, with a better prognosis usually associated with a later onset. Typical common clinical findings are central RPE atrophy, flecks and peripapillary sparing [91,96] (Figure 3).

Although ABCA4-retinopathy is caused by impaired function of a protein directly associated with vitamin A, there are only a few studies on the impact of vitamin A on its development [97]. According to dietary questionnaire studies and animal model studies, high vitamin A intake is thought to be a risk factor for the progression [76,98], therefore patients are discouraged from taking vitamin A supplements.

The vast majority of ongoing clinical trials (e.g., NCT03772665, NCT03364153, NCT02402660, NCT04239625) aim to target a particular step in the visual cycle and influence the process of A2E generation in various ways. However, these therapies are likely only to reduce symptoms of STGD1 and slower the rate of disease progression and not be curative [99,100]. An example is slowing the visual cycle by inhibiting RPE65 or RBP4, which deprives photoreceptors of 11-*cis*-retinal and consequently reduces the accumulation of toxic A2E. As a side effect, reversible night blindness occurs [101,102,103,104]. Alternative approach that does not compromise retinal function has been the provision of vitamin A deuterated at the carbon 20 position, which impedes the dimerization rate of vitamin A [105,106,107]. However, no therapy has yet been registered.

#### 2.2.2. Retinopathy Due to Pathogenic Variants in *RPE65*

*RPE65* gene is located on chromosome 1p31.3 and encodes 65 kDa enzyme. RPE65 catalyses the conversion of all-*trans*-retinyl ester to 11-*cis*-retinol, allowing it to be reused in phototransduction and preventing its conversion to potentially toxic molecules. Pathogenic variants in *RPE65* result in 11-*cis*-retinol and subsequently 11-*cis*-retinal deficiency, which essentially stops the visual cycle. In addition, retinyl esters accumulate in the RPE, leading to its damage [7,108,109,110]. The pathogenesis, therefore, combines local vitamin A deficiency and its impaired metabolism, which leads to the formation of toxic byproducts.

Patients with pathogenic variants in the *RPE65* gene have been presented with autosomal recessive RP, fundus albipunctatus, cone-rod dystrophy, and, most frequently, Leber congenital amaurosis (LCA) and early-onset severe retinal dystrophy (EOSRD) (LCA2, OMIM # 204100) [111]. Pathogenic variants in *RPE65* gene account for approximately 3% to 16% of LCA and EOSRD patients [112]. LCA and EOSRD are characterized by severe visual loss from birth or early infancy, wandering nystagmus, amaurotic pupils, and markedly reduced or non-recordable full-field electroretinograms [112,113]. Rod function is primarily affected, leading by cone dysfunction [114]. The fundus appearance is initially normal, followed by the development of pigmentary changes over time. OCT reveals thinning of the outer nuclear layer, while FAF images show low signal due to reduced lipofuscin accumulation in the RPE. Lipofuscin cannot form as a result of a non-functional visual cycle [115,116,117]. Most individuals are legally blind by the age of 40 [112].

Replacement therapy of missing 11-*cis*-retinol by synthetic 9-*cis*-retinyl acetate (QLT091001) has shown promising results in animal models, as it slowed the rate of retinal degeneration and improved the retinal function [118,119]. An open-label, prospective, multi-center, phase 1b clinical trial of oral administration of QLT091001 in patients with defective *RPE65* and *LRAT* genes also demonstrated improvement in vision. No serious adverse events occurred. However, some patients noted headaches, photophobia, nausea, reduction in serum high density lipoprotein concentrations, and increases in serum triglycerides, alanine aminotransferase and aspartate aminotransferase concentrations due to high doses of vitamin A [120,121].

These approaches have now been overshadowed by gene therapy for RPE65, that has been registered in 2017. The newly developed drug contains the active substance voretigene neparvovec. It employs adeno-associated virus vectors with genetically incorporated RPE65 cDNA. The introduction of a healthy *RPE65* gene into the retina restores the normal visual cycle, which improves vision and prevents or at least delays retinal degeneration [122,123,124].

#### 2.2.3. Retinitis Pigmentosa Due to Pathogenic Variants in *RHO*

Retinitis pigmentosa (RP) is the most common progressive hereditary retinal degeneration, affecting approximately 1 in 4000 people [125]. It is caused by pathogenic variants in almost 100 different genes and primarily affect rods. Since rods are responsible for low-light vision, night blindness is the first and characteristic manifestation of RP. In the later stage of the disease the cones are also affected, leading to visual field constriction and eventually central visual loss. Typical clinical findings, composing RP triad, are peripheral bone spicule pigmentation, attenuation of retinal vessels, and a waxy pallor of the optic nerve (Figure 4A) [125,126]. In most patients, FAF images show hyperautofluorescent ring, representing a transition zone between normal retinal function within the ring and absent outside of the ring. Hypoautofluorescent patches of RPE atrophy in the mid-periphery, related to a loss of peripheral vision, are normally seen in FAF images. Ring progressively constrict with time (Figure 4C). The OCT image shows the perifoveal loss of the outer retina and the central preservation of the ellipsoid zone, which corresponds to the internal edges of the hyperautofluorescent ring visible on FAF (Figure 4C1) [125].

RP can be inherited in different patterns including X-linked, autosomal recessive, and autosomal dominant [125,126]. The most common cause of autosomal dominant RP are pathogenic variants in the *RHO* gene that are responsible for 16% to 35% of cases with autosomal dominant RP [127].

The vast majority of pathogenic variants in the *RHO* gene have been identified in patients with autosomal dominant RP, while rare pathogenic variants are also known to cause autosomal recessive RP or autosomal dominant congenital stationary night blindness [127,128]. *RHO* was the first RP-linked gene identified. [129]. It is located on chromosome 3q22.1, encoding 39 kDa highly photosensitive G protein-coupled receptor in rods [126].

Clinical trial by Berson et al. reported that patients receiving 15,000 IU/d of vitamin A had a slower decline of 30-Hz ERG amplitude than those not receiving this dosage (*p* < 0.001), and therefore, retained more retinal function. Vitamin A was administered as retinyl palmitate. Patients were advised to maintain a regular diet without specifically selecting foods containing high levels of preformed vitamin A. The study included 601 patients (aged 18–49 years) with a treatment duration of 4 to 6 years [130]. Besides vitamin A, it was also shown that long-chain omega-3 fatty acids [131] and lutein intake slow the decline in visual function of patients with RP [132,133]. However, vitamin A, long-chain omega-3 fatty acids and lutein supplementation have not yet been established in clinical practice for treating patients with RP, as long-term safety and efficacy of lutein and long-chain omega-3 fatty acids are still object of research, whereas vitamin A supplementation may cause side-effects.

#### 2.2.4. Retinopathy Due to Pathogenic Variants in *RDH5* or *RDH11*

RDH5 and RDH11 are thought to be the main 11-*cis*-retinol dehydrogenases in the RPE. While RDH5 is responsible for most of the activity in the RPE, RDH11 plays a minor role [134].

*RDH5* gene locates on chromosome 12q13-q14 and encodes 32 kDa membrane-associated protein [135]. RDH5 catalyses the conversion of 11-*cis*-retinol to 11-*cis*-retinal through an NAD^+^-dependent reaction in the RPE, which is the final oxidation step in the visual cycle [136]. Pathogenic variants in *RDH5* have been associated with autosomal recessive fundus albipunctatus, which belongs to a group of hereditary flecked retina syndromes [137].

Fundus albipunctatus is a rare form of congenital stationary night blindness with delayed dark adaptation after exposure to bright light due to slowed production of 11-*cis*-retinal [138,139]. Impaired 11-*cis*-retinal production also leads to decreased A2E and lipofuscin formation, which results in reduced autofluorescence in FAF images [87,89,139,140,141]. It is characterised with stationary or slow progression of rod abnormalities, and numerous retinal flecks placed throughout the retina, except the fovea [139,142]. Flecks are thought to represent an accumulation of toxic retinyl esters in the RPE (Figure 5) [143]. Around 30% of patients with fundus albipunctatus, usually elderly, develop also progressive cone dystrophy [139,144].

Administration of 9-*cis*-β-carotene in a small group of patients with fundus albipunctatus showed promising results. However, no further studies on bigger cohorts were performed, and, therefore, 9-*cis*-β-carotene is not accepted as a treatment for patients with fundus albipunctatus [145].

Recovery of night vision after prolonged dark adaptation suggests that RDH11 plays a complementary role to RDH5 [143]. Besides participating in the oxidation of 11-*cis*-retinol to 11-*cis*-retinal, RDH11 also catalyzes a reduction of 11-*cis*-retinal. Moreover, it has an ability to catalyze the oxidation of *trans*-retinol and reduction of *trans*-retinal [60,134,146,147]. RDH11 has an approximate molecular mass of 35 kDa and is encoded by the *RDH11* gene located on chromosome 14q24.1. Pathogenic variants in the *RDH11* gene are associated with syndromic RP, which features include atypical RP, facial dysmorphologies, psychomotor developmental delay, learning disabilities and short stature [148]. Decreased FAF has also been described, as lipofuscin formation is reduced [87].

#### 2.2.5. Retinopathy Due to Pathogenic Variants in *RDH8* or *RDH12*

*RHD8* is located on chromosome 19p13.2, while *RDH12* locates on chromosome 14q24.1. RDH12, with a molecular mass of 35 kDa, and RDH8, with a molecular mass of about 34 kDa, are the major all-*trans*-retinal dehydrogenases in the photoreceptors and use NADPH as a cofactor [46,60]. RDH8 is located in the outer segment of cones and rods, while RDH12 is in the inner segments of cones and rods. RDH8 contributes to approximately 70% and RDH12 to approximately 30% of all-*trans*-RDH activity. Their dysfunction subsequently leads to decreased synthesis of 11-*cis*-retinal, which may result in reduced autofluorescence signal [46,47,60,149,150].

Pathogenic variants in *RDH12* have been shown to cause autosomal recessive LCA and EOSRD (LCA13, OMIM #612712), characterized by severe progressive rod-cone dystrophy with widespread RPE atrophy, bone-spicule pigmentation, vascular attenuation, macular atrophy and corresponding macular excavation [113,149,151,152]. A phenotypic feature of RDH12 retinopathy is sparing of the peripapillary region [149,151,153] (Figure 6), pathognomonic of ABCA4-retinopathy. Mutations in *RDH12* gene have been found in patients with autosomal recessive retinitis RP, cone-rod dystrophy as well as macular dystrophy [154]. Interestingly, pathogenic variants in *RDH8* have not yet been linked to any retinal disease.

#### 2.2.6. Retinopathy Due to Pathogenic Variants *RLBP1*

The retinaldehyde-binding protein 1 (*RLBP1*) gene is localized on chromosome 15q26 and transcribes RLBP1 also known as CRALBP, abundant in the RPE and Müller cells [47,155,156]. CRALBP is a 36 kDa water-soluble protein, that transports hydrophobic 11-*cis*-retinoids [48,49,157,158]. Pathogenic variants in the *RLBP1* gene prevent regeneration of 11-*cis*-retinal, which leads to reduced autofluorescence signal [87,159]. They have been associated with a number of autosomal recessive retinal dystrophies, including RP, fundus albipunctatus, Bothnia dystrophy, Newfoundland rod-cone dystrophy and retinitis punctata albescens [150,159,160]. The latter has been linked by pathogenic variants in five different genes, whereas the *RLBP1* gene is the most frequently associated with retinitis punctata albescens [161]. Retinitis punctata albescens is a rod-cone dystrophy subtype of RP and accounts for 0.5% of all RP cases, with a prevalence of 1 in 800,000 [162]. It is characterised by nyctalopia in early childhood and numerous punctate whitish-yellow spots throughout the entire fundus. Atrophy of the RPE appears mid-peripherally and progressively enlarge towards the centre. Macular involvement is frequent [156,160,162,163]. Dark adaptation is always abnormal and occurs even prior retinal degeneration [159,162]. RLBP1 gene therapy clinical trial is currently ongoing (NCT03374657).

#### 2.2.7. Retinopathy Due to Pathogenic Variants in *RBP3*

IRBP, also known as retinol binding protein 3 (RBP3), is a136 kDa lipoglycoprotein product of the *RBP3* gene on chromosome 10q11.22 [164,165]. It is secreted by photoreceptors and accumulates in the interphotoreceptor matrix. It functions as the transporter of retinoids between the photoreceptors and RPE and also between photoreceptors and Müller cells [29,50,66,166]. Pathogenic variants in *RBP3* gene are described as an infrequent cause of autosomal recessive RP [167]. They can also cause unusual retinal dystrophy characterized by childhood onset high myopia, generalized rod and cone dysfunction, and an unsignificant fundus appearance [168].

#### 2.2.8. Retinopathy Due to Pathogenic Variants in *RBP4*

RBP4 is a specific carrier for all-*trans*-retinol in the circulation, encoded by the *RBP4* gene, which is located on chromosome 10q23.33 [165]. To prevent extensive kidney filtration and lowered plasma concentrations of RBP4, holo-RBP4 associates with TTR. Therefore, the formation of holo-RBP:TTR complex represents an important step in vitamin A homeostasis [35,37]. Pathogenic variants in *RBP4* gene lead to reduced plasma RBP4 and all-*trans*-retinol concentrations and are associated with night blindness [34,169,170,171], microphthalmia [171,172], anophthalmia [172], coloboma [34,170,172], retinal dystrophy [34,169,170,171,173] and acne vulgaris [34,170,173].

#### 2.2.9. Retinopathy Due to Pathogenic Variants *RGR*

*RGR* gene is located on chromosome 10q23.1 [174] and encodes an 28 kDa intracellular opsin localized to RPE and Müller cells [175,176]. As the association of variants in *RGR* gene with specific ocular diseases has been rarely reported [55,56], clinical data on are very limited. Spectrum of retinal changes in published cases ranged from normal visual acuity, normal electroretinogram and diffuse or reticular pigmentation of the retina to severe visual loss with diffuse atrophy of the retina and choroid [177].

#### 2.2.10. Retinopathy Due to Pathogenic Variants in *LRAT*

LRAT is a 36 kDa protein encoded by *LRAT* gene located on chromosome 4q32.1 [178]. It converts all-*trans*-retinol to all-*trans*-retinyl esters. Pathogenic variants in *LRAT* gene cause LCA and RP [45,179,180]. Similarly to pathogenic variants in the *RPE65* gene, pathogenic variants in *LRAT* also lead to 11-*cis*-retinal deficiency and, therefore, lack of autofluorescence [87,180]. Treatment with oral synthetic *cis*-retinoid QLT091001 showed promising results in patients. However, the adverse effect always have to be a concern [120,121].

#### 2.2.11. Retinopathy Due to Pathogenic Variants *STRA6*

*STRA6* gene is located on chromosome 15q24.1 and encodes transmembrane receptor [181], with a molecular mass of 74 kDa [182]. It catalyses the release of retinol from RBP4 and facilitates its translocation across the RPE cell membrane to the cytosol [41,181]. *STRA6* gene pathogenic variants are associated with Matthew-Wood syndrome, which is in eyes presented with anophthalmia [47,181,183,184,185,186].

### 2.3. Retinal Diseases Involving Local Vitamin A Deficiency

#### 2.3.1. Sorsby Fundus Dystrophy TIMP3

Sorsby fundus dystrophy (SFD) is a rare retinal dystrophy with autosomal dominant pattern of inheritance. The estimated prevalence is 1 in 220,000 [187]. It is caused by pathogenic variants in the gene encoding a tissue inhibitor of metalloproteinases-3 (TIMP3), located on chromosome 22q12.3 [188] which encodes a 24 kDa glycoprotein [187]. TIMP3 is mainly expressed and secreted by the RPE and is an element of Bruch’s membrane. It regulates turnover of the extracellular matrix, inflammation and possesses pro-apoptotic and anti-angiogenic activities [77,189].

Pathogenic variants in the *TIMP3* gene result in thickening of the Bruch’s membrane, leading to reduced exchange of nutrients and waste products between RPE and choriocapillaris, including vitamin A, resulting in local vitamin A deficiency [77,187]. Symptoms usually appear after the 2nd decade of life, with an average onset in the 4th to 6th decade [190]. Typically, the earliest symptom is night blindness. However, later the disease is often complicated with choroidal neovascularizations, leading to metamorphopsia, reduced colour vision, and loss of central vision. Clinically, the disease manifests with thickened Bruch’s membrane, drusen-like deposits in the sub-RPE space, and choroidal neovascularization, which usually requires treatment with vascular endothelial growth factor (VEGF) inhibitors. Later, expanding areas of geographic atrophy may also develop [77,187,189,191,192,193] (Figure 7). Indocyanine green angiography can show blockade of choroidal fluorescence by the thickened Bruch membrane [189,194] (Figure 7E).

Early attempts of SFD treatment included adding vitamin A to improve symptoms of night blindness. Although initial results were promising, treatment did not become widely used due to the potential long-term toxicity of high doses of vitamin A and lack of lower doses efficacy [77,189].

#### 2.3.2. Age-Related Macular Degeneration

AMD is the most common cause of blindness in the elderly, with the risk increasing with age. It is a complex, multi-factorial disease that is affected by both genetic factors as well as environmental factors like ageing itself, smoking and obesity [7,195].

In AMD, there is a marked thickening of the Bruch’s membrane, which reduces the permeability of nutrients and waste products through it [196,197]. As a result, waste material in the form of druses, consisting of cell residues, lipids, proteins and carbohydrates, is deposited between the RPE and the Bruch’s membrane [198,199].

With the thickening of the Bruch’s membrane, the access of vitamin A to the RPE and consequently photoreceptors is reduced, which causes similar consequences as in hypovitaminosis A [196,200]. Studies using scotopic microperimetry suggested that rod dysfunction may be present early on in disease [201]. One subtype of disease is characterized by co-called reticular drusen, which have been interestingly also observed in hypovitaminosis A [202] (Figure 8).

In a major clinical Age-Related Eye Disease Study (AREDS) the effect of nutritional supplements on the development and progression of AMD was evaluated. The original AREDS formula included beta-carotene, which was found to increase the risk of lung cancer in smokers. Therefore, in a new, improved AREDS2 formula, beta-carotene was replaced with lutein and zeaxanthin. Regular use of AREDS2 has been shown to reduce the risk of developing severe AMD [21,22,23].

Currently, two clinical studies with AMD patients are underway involving supplementation of vitamin A in the form of retinyl palmitate (NCT03478865, NCT03478878).

## 3. Summary of the Relevant Clinical Trials

There are many different preclinical and clinical trials where researchers try to elucidate the mechanisms and develop effective treatments for retinal diseases associated with vitamin A. Some of them include targeted gene therapy, while others include non-genetic therapies affecting the visual cycle. The most relevant trials for each disease are stated in each chapter above and summarized in Table 2.

## 4. Conclusions

Normal level of vitamin A is essential for good vision, while either too much and too little can be harmful. Vitamin A needs to be supplemented in night blindness due to vitamin A deficiency but should be avoided in STGD1, where toxic products of vitamin A accumulate excessively. Despite its known involvement in various other retinal diseases, reviewed herein, routine administration of vitamin A is currently not advised for those patients. This illustrates the need for further studies involving vitamin A metabolism and the effect of treatment with vitamin A and/or other related compounds in these patients. Currently, more than 100 different studies are underway on this topic, which we hope will provide answers on the mechanisms and treatment of retinal diseases associated with vitamin A metabolism.

## Figures and Tables

**Figure 1 ijms-23-01014-f001:**
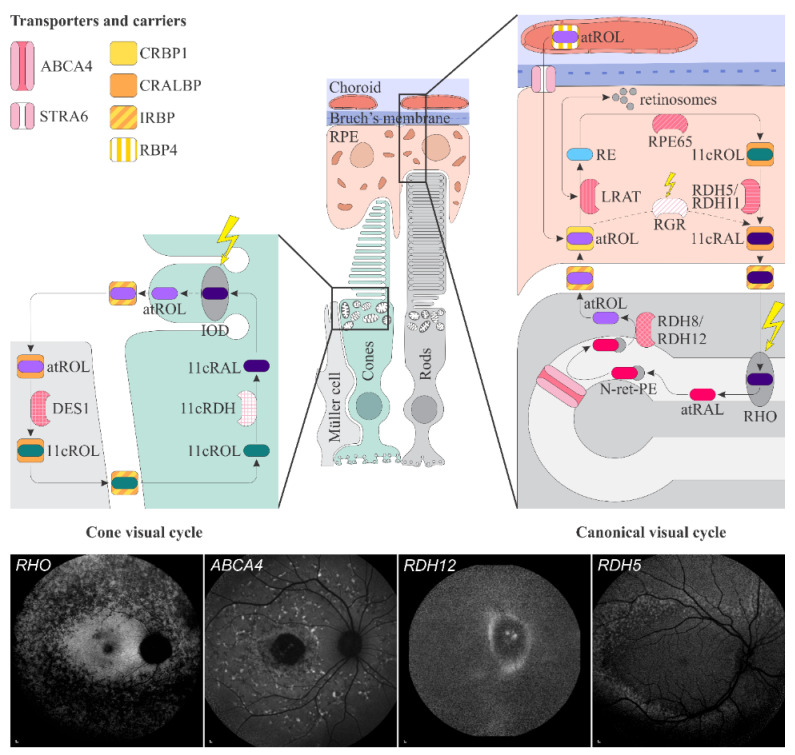
Schematic representation of the canonical and cone visual cycles. Key enzymes, transporters, carriers and retinoids are shown. Fundus autofluorescence (FAF) images of representative patients, harbouring pathogenic variants in genes, encoding four essential proteins involved in the visual cycle, are presented below. Scale bars for four FAF images: 200 µm. Abbreviation expanation: RPE–retinal pigment epithelium, RBP4—retinol-binding protein 4, STRA6—stimulated by retinoic acid 6, CRBP1—cellular retinol-binding protein 1, LRAT—retinol:lecithin acyltransferase, RPE65—RPE-specific 65 kDa protein, RDH5—11-*cis*-retinol dehydrogenase 5, RDH11—11-*cis*-retinol dehydrogenase 11, CRALBP—cellular retinaldehyde-binding protein, RGR—retinal G protein-coupled receptor, IRBP—interphotoreceptor retinoid-binding protein, RDH8—all-*trans*-retinol dehydrogenases 8, RDH12—all-*trans*-retinol dehydrogenases 12, ABCA4—ATP-binding cassette subfamily A member 4, DES1—dihydroceramide desaturase-1, 11cRDH—11-*cis*-retinol dehydrogenase, RHO—rhodopsin, IOD—iodopsin, 11cRAL—11-*cis*-retinal, 11cROL—11-*cis*-retinol, atRAL—all-*trans*-retinal, atROL—all-*trans*-retinol, RE—retinyl esters, N-ret-PE—*N-retinylidene-phosphatidylethanolamine*, PE—*phosphatidylethanolamine*, A2E—*N-retinyl-N-retinylidene ethanolamine*. Source: Eye Hospital, University Medical Centre Ljubljana.

**Figure 2 ijms-23-01014-f002:**
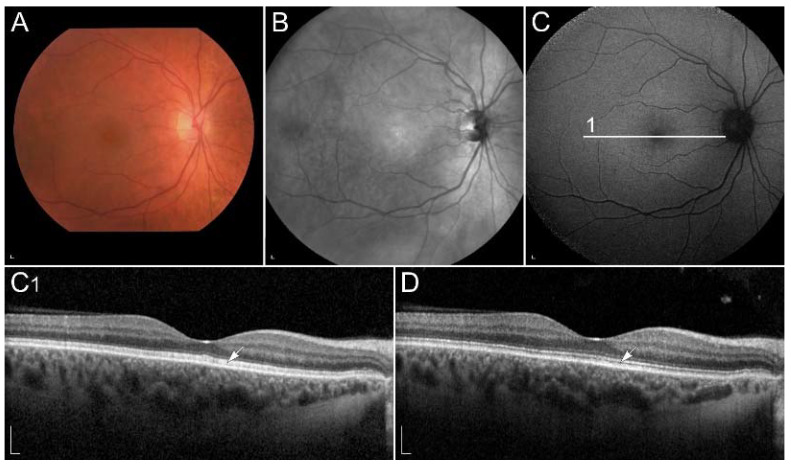
(**A**) Colour fundus image, (**B**) fundus infrared image and (**C**) FAF image in a patient with vitamin A deficiency. Note the corresponding (**C1**, arrow) spectral-domain optical coherence tomography (SD-OCT) image with abnormalities in photoreceptor outer segments, which (**D**, arrow) normalized after treatment with vitamin A supplementation. Patient’s best corrected Snellen decimal visual acuity before treatment was 0.7 on the right eye and 0.6 on the left eye. After treatment visual acuity improved to 1.0 on the right eye and 0.9 on the left eye. Scale bars: 200 µm. Source: Eye Hospital, University Medical Centre Ljubljana.

**Figure 3 ijms-23-01014-f003:**
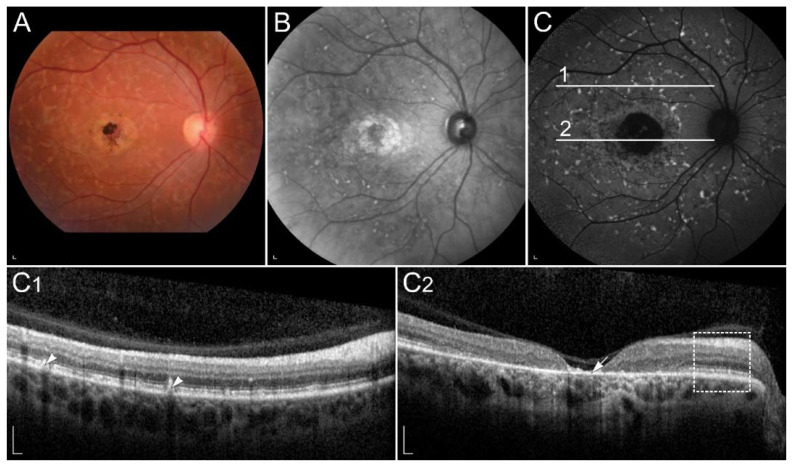
Clinical findings in a patient harbouring p.(Trp431*) and p.(Asp262Gly) in *ABCA4*. Mac-ular affection, fundus flecks and peripapillary sparring (diagnostic triad) are shown on (**A**) colour fundus image, (**B**) fundus infrared image and (**C**) FAF image. On corresponding (**C1**,**C2**) SD-OCT images, (arrowheads) hyperautofluorescent flecks, (arrow) RPE atrophy and (rectangle) peripapillary sparing can also be observed. Patient’s best corrected Snellen decimal visual acuity was 0.1 on both eyes. Scale bars: 200 µm. Source: Eye Hospital, University Medical Centre Ljubljana.

**Figure 4 ijms-23-01014-f004:**
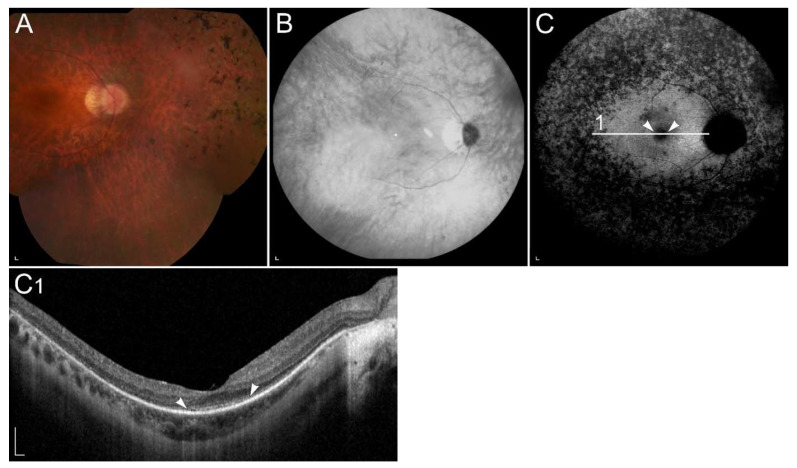
(**A**) Colour fundus image, (**B**) fundus infrared image and (**C**) FAF image with corresponding (C1) SD-OCT image showing clinical findings in a patient with p.(Gly90Asp) in *RHO*. Area within the arrowheads on C and C1 images corresponds to the preserved part of the retina. Patient’s best corrected Snellen decimal visual acuity was 0.6 on the right eye and 0.4 on the left eye. Scale bars: 200 µm. Source: Eye Hospital, University Medical Centre Ljubljana.

**Figure 5 ijms-23-01014-f005:**
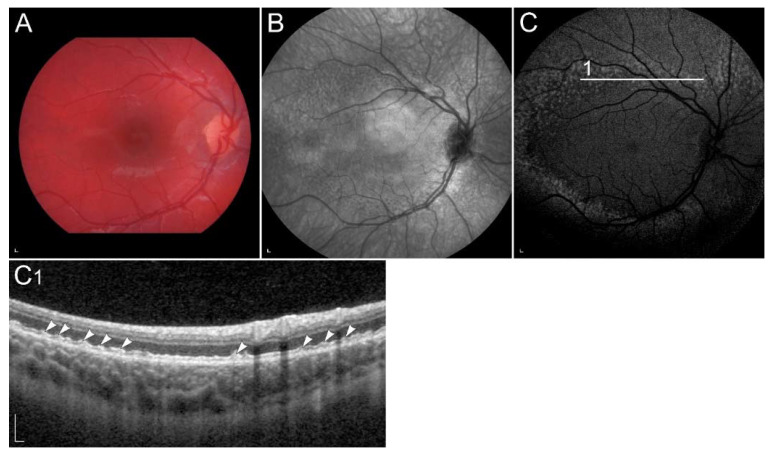
Clinical characteristics of a patient homozygous for p.(Thr137Ser) in *RDH5*. (**A**) Colour fundus image, (**B**) infrared image and (**C**) FAF image with corresponding (**C1**) SD-OCT image showing (arrowheads) retinal flecks. FAF image shows reduced autofluorescence in the entire retina. Patient’s best corrected Snellen decimal visual acuity was 0.8 on both eyes. Scale bars: 200 µm. Source: Eye Hospital, University Medical Centre Ljubljana.

**Figure 6 ijms-23-01014-f006:**
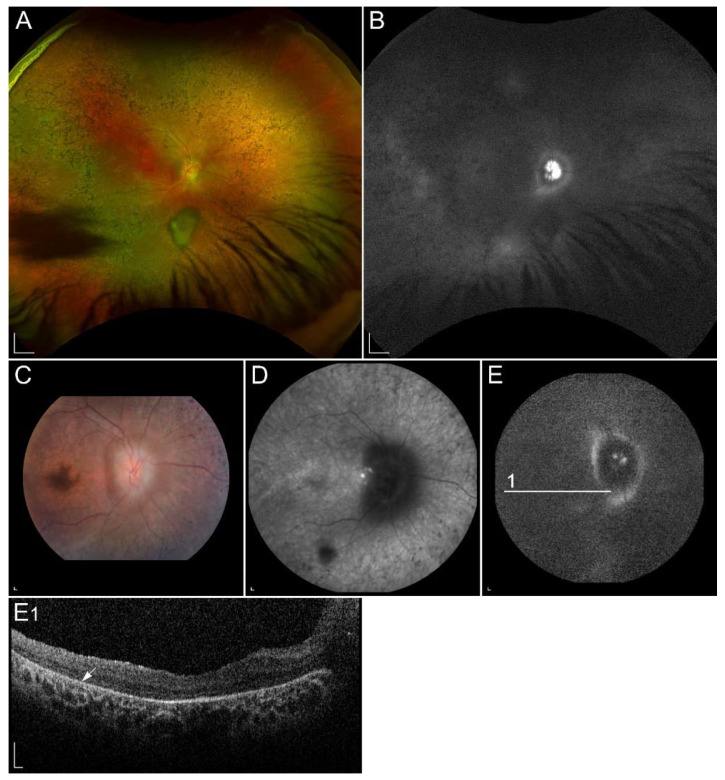
(**A,C**) Ultra-widefield and 50° colour fundus images, (**D**) fundus infrared image, (**B,E**) ul-tra-widefield and 55° FAF images and (**E1**) SD-OCT image of a patient homozygous for p.(Ala126Glu) in *RDH12*. (**A,C**) Colour fundus images and (**D**) fundus infrared image demonstrate bone-spicule pigmentation and wide-spread retinal atrophy. The latest is also shown on (**E1**) OCT, which shows loss of the photoreceptor inner segment ellipsoid band and (arrow) atrophy of the RPE. Peripapillary retinal preservation can be seen on A, B, C, D and E images. The quality of E and E1 images is low due to reduced visual acuity and nystagmus. Patient’s best corrected Snellen decimal visual acuity was 0.3 on both eyes. Scale bars A, B: 2 mm; C, D, E and E1: 200 µm. Source: Eye Hospital, University Medical Centre Ljubljana.

**Figure 7 ijms-23-01014-f007:**
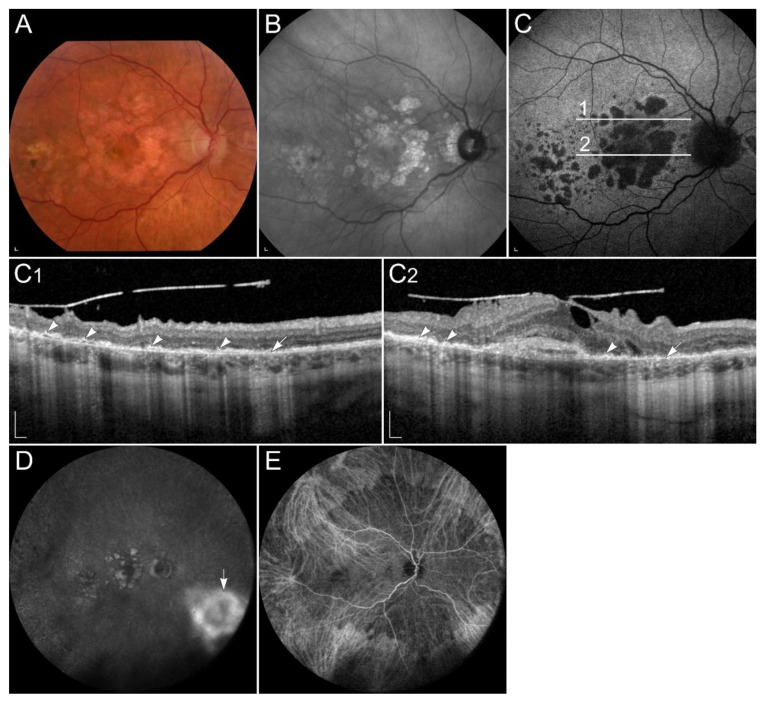
Patient with Sorsby dystrophy due to pathogenic variant p.(Ser170Cys) in *TIMP3*. (**A**) Colour fundus image, (**B**) fundus infrared image, and (**C**) FAF image with corresponding (**C1,C2**) SD-OCT images showing (arrows) geographic atrophy and (arrowheads) subretinal drusen-like deposits. (**D**, arrow) Hyperfluorescent area is compatible with choroidal neovascularisation. (**E**) The reduced indocyanine green late-phase angiography fluorescence due to decreased permeability of Bruch’s membrane. Patient’s best corrected Snellen decimal visual acuity was 0.4 on the right eye and 0.6 on the left eye. Scale bars: 200 µm. Source: Eye Hospital, University Medical Centre Ljubljana.

**Figure 8 ijms-23-01014-f008:**
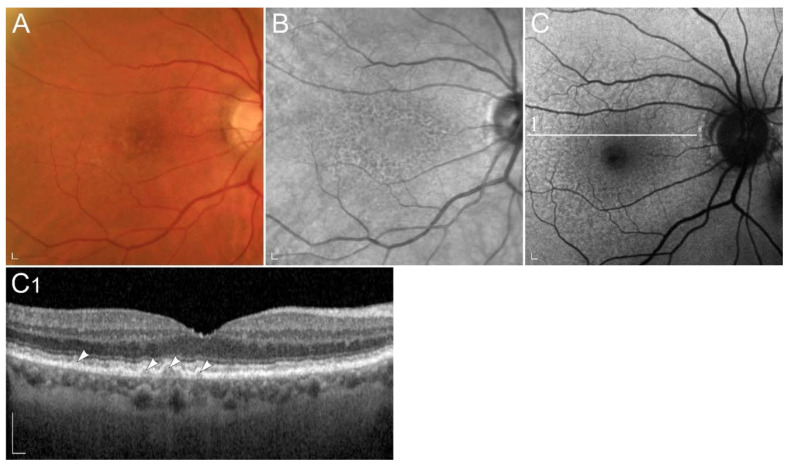
Patient with age related macular degeneration. Representative (**A**) colour fundus image, (**B**) fundus infrared image and (**C**) FAF image with corresponding (**C1**) SD-OCT image show (arrowheads) reticular drusen. Patient’s best corrected Snellen decimal visual acuity was 0.8 on both eyes. Scale bars: 200 µm. Source: Eye Hospital, University Medical Centre Ljubljana.

**Table 1 ijms-23-01014-t001:** Summary of roles and pathogenetic mechanisms in diseases associated with vitamin A metabolism.

Gene (Protein)	Location	Role	Associated Disease	Proposed Pathogenic Mechanism
*ABCA4* (ABCA4)	1p22.1	Transport of retinoids to the cytoplasmic membrane of rods and cones.	STGD1	Accumulation of toxic retinoids in the RPE, direct cone toxicity.
*LRAT* (LRAT)	4q32.1	Conversion of all-*trans*-retinol to all-*trans*-retinyl esters	LCA, RP	11-*cis*-retinol deficiency, reduced lipofuscin accumulation in the RPE.
*RBP1* (CRBP1)	3q23	Transport of all-*trans*-retinol in the retina.	No human retinal diseases have been associated with pathogenic variants in *RBP1* gene.	/
*RBP3* (IRBP)	10q11.22	Transport of retinoids between photoreceptors, RPE and Müller cells.	RP, unusual retinal dystrophy	Disabled protection and solubilization of visual cycle retinoids.
*RBP4* (RBP4)	10q23.33	Transport of all-*trans*-retinol in the crculation and its delivery to the STRA6.	Night blindness, microphthalmia, anophthalmia, coloboma, retinal dystrophy, acne vulgaris	Reduced plasma concentrations of RBP4 and all-*trans*-retinol.
*RDH5* (RDH5)	12q13-q14	Conversion of 11-*cis*-retinol to 11-*cis*-retinal.	Fundus albipunctatus	Slowed production of 11-*cis*-retinal, reduced lipofuscin accumulation in the RPE.
*RDH8* (RDH8)	19p13.2	Conversion of all-*trans*-retinal to all-*trans*-retinol.	Pathogenic variants are not linked to any retinal disease	/
*RDH11* (RDH11)	14q24.1	Oxidation and reduction of *cis*-retinoids and *trans*-retinoids.	Syndromic RP	Slowed production of 11-*cis*-retinal, reduced lipofuscin accumulation in the RPE.
*RDH12* (RDH12)	14q24.1	Conversion of all-*trans*-retinal to all-*trans*-retinol.	LCA	Decreased synthesis of 11-*cis*-retinal, reduced lipofuscin accumulation in the RPE.
*RGR* (RGR)	10q23.1	Contributes to the regeneration of 11-*cis*-retinal.	Association with specific ocular diseases has been rarely reported	Decreased synthesis of 11-*cis*-retinal.
*RHO* (RHO)	3q22.1	G protein-coupled photosensitive receptor in rods	Congenital stationary night blindness, RP	Dysfunction in phototransduction.
*RLBP1* (CRALBP)	15q26	Transports of 11-*cis*-retinoids in RPE and Müller cells.	RP, fundus albipunctatus, Bothnia dystrophy, Newfoundland rod-cone dystrophy and retinitis punctata albescens.	Regeneration of 11-*cis*-retinal is prevented, reduced lipofuscin accumulation in the RPE.
*RPE65* (RPE65)	1p31.3	Conversion of all-*trans*-retinal to 11-*cis*-retinal.	LCA and EOSRD (LCA2), RP, fundus albipunctatus, cone-rod dystrophy	11-*cis*-retinol deficiency, leading to visual cycle interruption, reduced lipofuscin accumulation in the RPE, retinyl esters accumulation in the RPE.
*STRA6* (STRA6)	15q24.1	Release of retinol from RBP4 and its translocation across the RPE	Matthew-Wood syndrome	Severe reduction of vitamin A transport into RPE cells.
*TIMP3* (TIMP3)	22q12.3	Regulation of extracellular matrix turnover, inflammation, pro-apoptotic and anti-angiogenic activities.	SFD	Impaired transfer of vitamin A through the thickened Bruch’s membrane.

Abbreviation explanation: ABCA4—ATP-binding cassette subfamily A member 4, STGD1—Stargardt disease, RPE—retinal pigment epithelium, RPE65—RPE-specific 65 kDa protein, LCA—Leber congenital amaurosis, EOSDR—early-onset severe retinal dystrophy, RP—retinitis pigmentosa, RHO—rhodopsin, RDH5—11-*cis*-retinol dehydrogenase 5, RDH11—11-*cis*-retinol dehydrogenase 11, RDH8—all-*trans*-retinol dehydrogenases 8, RDH12—all-*trans*-retinol dehydrogenases 12, CRALBP—cellular retinaldehyde-binding protein, RLBP1—retinaldehyde-binding protein 1, CRBP1—cellular retinol-binding protein 1, RBP1—retinol binding protein 1, IRBP—interphotoreceptor retinoid-binding protein, RBP3—retinol binding protein 3, RBP4—retinol-binding protein 4, RGR—retinal G protein-coupled receptor, LRAT—retinol:lecithin acyltransferase, STRA6—stimulated by retinoic acid 6, TIMP3—tissue inhibitor of metalloproteinases-3, SFD—Sorsby fundus dystrophy.

**Table 2 ijms-23-01014-t002:** Summary of the relevant clinical trials.

NCT Number	Disease	Drug	Sponsor	Number of Subjects	Phase of the Study	Mechanism
NCT03772665	STGD1	Emixustat (inhibitor of RPE65)	Kubota Vision Inc.	194	3	Slower regeneration of 11-*cis*-retinal.
NCT03364153	STGD1	Zimura (anti-C5 aptamer)	IVERIC bio, Inc.	120	2	Prevention of the destructive effects of the activated complement cascade.
NCT02402660 and NCT04239625	STGD1	ALK-001 (C20-deuterated vitamin A)	Alkeus Pharmaceuticals, Inc.	140	2	Impaired dimerization of vitamin A and therefore reduced production of A2E.
NCT03374657	RP	CPK850 (*RLBP1* promoter)	Novartis Pharmaceuticals	Recruiting	1 and 2	Gene therapy.
NCT03478865 and NCT03478878	AMD	Vitamin A palmitate	National Eye Institute (NEI)	Recruiting	1	Vitamin A supplementation.

Abbreviation explanation: AMD–age-related macular degeneration. Source: ClinicalTrials.gov, accessed on 7 January 2022.

## Data Availability

The data presented in this study are available on request from the corresponding author. The data are not publicly available due to personal data protection.

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
