# Peer review of "The Role of Vitamin A in Retinal Diseases"

_ijms, 2022, doi:10.3390/ijms23031014_

Round 1

Reviewer 1 Report

The manuscript provides an excellent and comprehensive review of retinopathies directly or indirectly associated with Vitamin A. In general, the manuscript is very well written with appropriate references to existing literature including the most recent ones. The figures are relevant, appropriate, and readily interpretable. Below I outline few minor concerns that the authors can consider and address when possible. I believe this will improve the overall impact of the work.

General Comments:

  • Best corrected visual acuity or visual acuity levels has not been included in any section of the manuscript. Including this information, say in the figure legend of the 7 illustrative figures could help the reader understand the visual impact of retinopathies.
  • As authors rightly point out several clinical trials are either underway or have been completed that are related to Vit A/ visual cycle. Perhaps a summary table can list the trials (say NCT numbers, sponsor, number of subjects, phase of the study and/or mechanism). This can serve as an easy reference for the readers to follow the potential therapies.

Specific Comments:

  • Line 60: The order of the words in this sentence seems to be jumbled.
  • Line 250 – Table 1: Can be sorted either alphabetically by protein or by retinal localization (anterior to posterior)
  • Table 1 – For most cases the protein and gene info are same and so is redundant. Instead, it might be worthwhile to supply the chromosome loci of the gene.
  • Line 294: “150 different studies” – Is this an estimate, if yes say that explicitly. If not please cite the source.
  • Line 320, Figure 2 and elsewhere: Using arrows or other markers point out the abnormalities in the outer segment for the OCT imaging.
  • Line 431: Perhaps include markers on both FAF and OCT to mark this island of residual vision.
  • Line 452: Consider including a sentence or two to expand as to why supplementation was not adopted in clinical practice. ??side-effects ??risk outweigh benefits
  • Fig 6E and 6E1 – Image quality seems to be relatively low when compared to other, suggesting that the acuity was severely impaired/fixation was poor. Can include that if that is true.

Reviewer 2 Report

The present review summarises the clinical features of different retinal diseases that are caused by a dysfunction in the vitamin A pathway. It also gives a comprehensive overview on the underlining pathology. Through a discussion of a role of vitamin A in different retinal disorders, the manuscript highlights the importance of vitamin A in maintaining vision. A few published reviews discuss the role of vitamin A in retinal diseases however they tend to focus on a particular disorder such as retinitis pigmentosa. The review should add values to the relevant research field. While it is informative to present images illustrating the clinical features of the highlighted retinal diseases, it is necessary to cite the sources of the images in the legend (Figures 2 to 8). The authors should limit critical analysis to published findings. Lines 505 to 509 appear to discuss the differences between author’s new data (Figure 6) and literatures, which should be avoid in a review.
